# Advances in Local Drug Delivery for Periodontal Treatment: Present Strategies and Future Directions

**DOI:** 10.3390/biom15060903

**Published:** 2025-06-19

**Authors:** Mayuka Nakajima, Mayuko Yanagawa, Honoka Takikawa, Truong Tran Thien, Lorena Zegarra-Caceres, Chunyang Yan, Koichi Tabeta

**Affiliations:** Division of Periodontology, Faculty of Dentistry & Graduate School of Medical and Dental Sciences, Niigata University, Niigata 951-8514, Japan; yanagawa@dent.niigata-u.ac.jp (M.Y.); takikawa@dent.niigata-u.ac.jp (H.T.); truongtran@dent.niigata-u.ac.jp (T.T.T.); lorena@dent.niigata-u.ac.jp (L.Z.-C.); chunyang@dent.niigata-u.ac.jp (C.Y.)

**Keywords:** periodontitis, local drug delivery systems, nanotechnology, antibiofilm therapy, bone regeneration

## Abstract

Periodontitis is a highly prevalent, irreversible inflammatory disease characterized by the destruction of tooth-supporting tissues, eventually leading to tooth loss. Conventional treatment involves the mechanical removal of the subgingival biofilm, which is a major cause of gingival inflammation. However, the inaccessibility of deep-seated polymicrobial biofilms limits its effectiveness. Despite the adjunct use of systemic antimicrobials, their low site-specific bioavailability and systemic side effects remain concerns. Local drug administration offers a targeted alternative. However, the dynamic oral environment, which is characterized by continuous salivary and gingival crevicular fluid flow, poses challenges in maintaining therapeutic drug levels. Drug delivery systems (DDSs) provide technical solutions to overcome these limitations. With advancements in materials science and nanotechnology, diverse local DDS (LDDS) formulations tailored for periodontal applications have been developed. While traditionally focused on infection control, the application of LDDSs has expanded beyond antimicrobial therapy. Increasing attention has been paid to LDDS-based regenerative strategies, which aim to overcome the limitations of conventional regenerative therapies. This review aims to provide a comprehensive overview of the current and emerging DDS strategies in periodontal therapy, focusing on their applications in infection management and bone regeneration and discussing their limitations and prospects for clinical translation.

## 1. Introduction

Periodontitis is a highly prevalent, irreversible chronic inflammatory disease that destroys tooth-supporting tissues and can lead to tooth loss. It significantly affects oral function, esthetics, and overall quality of life [1]. Additionally, periodontitis is associated with diabetes and is a potential risk factor for other systemic diseases, such as cardiovascular disease and rheumatoid arthritis [2,3]. Severe periodontitis affects approximately 19% of the global adult population, accounting for over 1 billion cases worldwide [4]. Given its health burden and global prevalence, the development of effective therapeutic strategies is a major public health priority.

Periodontitis is caused by pathogenic microorganisms that colonize the periodontal sulcus, the space between the tooth and gingival tissue, with polymicrobial biofilms on the tooth surfaces [5]. These biofilms promote bacterial persistence and resist immune responses and antimicrobial agents [6]. The accumulation of these biofilms provokes a sustained host immune response, progressively leading to connective tissue damage with the deepened gingival sulcus as a periodontal pocket, resulting in alveolar bone loss and eventual tooth loss.

Therefore, the elimination of biofilms is a critical strategy in periodontal therapy for addressing the underlying cause of the disease. Mechanical debridement, such as scaling and root planing (SRP), is the standard approach. However, mechanical debridement alone cannot completely eradicate pathogens due to the limited accessibility of biofilms residing deep within the periodontal pockets and the anatomical complexities of the tooth [7]. Therefore, adjunctive antimicrobial therapy has been introduced. Systemic antimicrobial therapy is widely used. However, it has inherent drawbacks, including reduced bioavailability at the disease site due to first-pass hepatic metabolism and potential systemic side effects [8]. Therefore, clinicians often employ local drug administration to the inflamed sites to overcome these limitations. However, the oral cavity is a highly dynamic environment characterized by a continuous saliva and gingival crevicular fluid (GCF) flow, which poses significant challenges in maintaining therapeutic drug concentrations for a sufficient duration [9]. Therefore, developing strategies to overcome these challenges is crucial.

Drug delivery systems (DDSs) provide technical solutions to overcome these limitations by enabling a controlled, sustained, and localized drug release. With advancements in materials science and nanotechnology, the development of DDS formulations tailored for periodontal applications, primarily targeting infection control, has been boosted [10].

The management of periodontal infection enhances alveolar bone regeneration, which is the ideal goal of periodontal therapy. Although regenerative materials, such as bone grafts and growth factors, are routinely employed in periodontal surgery, the extent of bone regeneration achieved remains limited [11]. The application of DDSs for bone regeneration has attracted research interest and holds significant potential for future clinical translation [12].

Therefore, this review aims to provide a comprehensive overview of DDSs in periodontal therapy, covering currently available products and potential future candidates such as smart hydrogels, nanoparticles, and ionic liquids (ILs), with a focus on their applications in infection control and bone regeneration therapy. Additionally, this review emphasizes the therapeutic rationale and clinical significance behind each system, clarifying how each LDDS platform is designed to address specific unmet challenges in periodontal therapy. Finally, we also discuss the challenges and future perspectives of DDSs in this field.

## 2. Overview of DDSs in Periodontal Treatment

DDSs in periodontal therapy aim to maintain effective drug concentrations within the subgingival pockets over a sustained period. Local drug delivery systems (LDDSs) allow the direct application of therapeutics to affected sites because of the anatomical accessibility of periodontal tissues. LDDSs improve the local bioavailability, reduce the dosage and administration frequency, minimize systemic side effects, and enable the use of agents unsuitable for systemic delivery, such as chlorhexidine (CHX) [13]. LDDSs should meet several key criteria to achieve optimal therapeutic efficacy [10], including (1) ease of application in narrow, deep periodontal pockets, (2) strong retention against saliva and GCF, (3) a controlled and sustained drug release to prevent reinfection, (4) biocompatibility, and (5) biodegradability.

The materials that have been used in clinical LDDS applications can be categorized into fibers, strips and films (SFs), microparticles (MPs), and gels [14] (Figure 1). Each type presents distinct advantages and limitations, which will be reviewed in the following section, with reference to their therapeutic performance and how they fulfill the ideal LDDS criteria (Table 1). Additionally, recent advancements in LDDS technologies to overcome limitations and expand their therapeutic potential, including novel DDS strategies for alveolar bone regeneration, will be explored in the future perspectives section.

## 3. Current Status of LDDSs in Periodontal Therapy

### 3.1. Fibers

Fibers are a reservoir type of DDS designed for placement around the circumference of the periodontal pocket using an applicator, providing excellent retention within the pocket [15]. Fibers, which are typically composed of nonabsorbable polymers, incorporate antimicrobial agents through surface adsorption or blending with the polymer matrix, allowing for a controlled and sustained drug release [16,17].

Actisite^®^ (Kalamazoo, MI, USA) was the first tetracycline hydrochloride-loaded fiber approved by the United States Food and Drug Administration (USFDA). It was introduced in 1994 for periodontal therapy. Actisite^®^ is composed of ethylene–vinyl acetate and maintains effective antibacterial concentrations (>1000 μg/mL) in the periodontal pocket for up to 10 days [18]. Despite its clinical benefits, Actisite^®^ was discontinued due to its nonbiodegradable nature, and a second visit for fiber removal was required, often leading to patient discomfort, gingival redness, and delayed healing [19].

Various biodegradable polymers, such as chitosan and zein, and synthetic alternatives, such as poly(lactide-co-glycolide) (PLGA) and poly(ε-caprolactone) (PCL), have been explored for fiber-based periodontal DDSs to overcome these limitations [20,21]. However, these absorbable fibers exhibit limited mechanical stability and retention in periodontal pockets. Furthermore, fiber placement is technically demanding and time-consuming, which poses challenges even for experienced clinicians. These practical limitations have hindered the widespread clinical adoption of fiber-based DDSs [22]. Therefore, no fiber-based system has been clinically available since the discontinuation of Actisite^®^.

### 3.2. Strips and Films

SFs are thin polymeric matrices in which therapeutic agents are uniformly dispersed. Adaptability is a key advantage of SFs, as it allows them to conform to the anatomy of periodontal pockets and facilitates easy insertion with minimal patient discomfort [23]. Early SF systems were composed of nonbiodegradable materials. However, biodegradable alternatives, such as those using poly-hydroxybutyric acid and PLGA, have become mainstream [24,25]. These biodegradable SFs enable sustained drug release through diffusion or polymer erosion mechanisms.

Advanced designs, such as bilayer films incorporating mucoadhesive chitosan and biodegradable polymers, have been developed to enhance retention within the periodontal pocket [26]. PerioChip^®^ (Or Akiva, Haifa District, Israel) is a USFDA-approved biodegradable gelatin-based chip loaded with chitosan. It provides prolonged drug release due to its inherent adhesiveness, with minimal patient discomfort [27]. Approximately 40% of the drug is released within the first 24 h, followed by sustained release over 7 days, maintaining concentrations above the minimum inhibitory concentration for common periodontal pathogens [28]. When used as an adjunct to SRP, PerioChip^®^ has been reported to significantly improve periodontal clinical parameters, including the gingival index (GI), probing pocket depth (PPD), and bleeding on probing (BOP), at 3 months, with further gains in clinical attachment levels (CAL) at 6 months.

Despite their crucial role in enhancing periodontal therapy, SFs are often challenged by burst drug release. Consequently, the emergence of micro- and nano-scale DDSs has shifted research efforts toward MPs, NPs, and gels as more advanced approaches for localized drug delivery [14].

### 3.3. Gels

Gels are semisolid systems in which a liquid phase is entrapped within a cross-linked polymeric network. Gels offer high biocompatibility, ease of injectable administration, and a relatively simple fabrication process, making them an attractive platform for periodontal applications [29]. The gels are classified into oleogels (oil-based), hydrogels (water-based), and bigels (combining both phases for dual delivery functionality).

Elyzol 25% (London, UK) dental gel, a commercially available oleogel in the UK, is formulated as a suspension of glycerol monooleate and sesame oil, with metronidazole benzoate being the active pharmaceutical ingredient. It provides a sustained drug release over 24–36 h and has demonstrated a significant reduction in PPD when used as an adjunct to SRP [30]. However, the poor biodegradability of oleogels is a key limitation that may affect their long-term biocompatibility [31].

Hydrogels have gained considerable attention due to their superior biocompatibility. Numerous studies have investigated their application in periodontal therapy. Chlosite^®^ (Casalecchio di Reno, Bologna, Italy) is a commercially available xanthan-based hydrogel loaded with CHX. It has demonstrated clinical benefits, including improvements in PPD and CAL, when used adjunctively with SRP [32,33]. Despite their potential, many hydrogel-based LDDSs for periodontitis are still in the experimental or early clinical stages. This is due to challenges such as the initial burst drug release and limited in vivo stability resulting from rapid clearance through normal catabolic pathways [34].

With recent advancements in polymer science, innovative gel formulations have been developed to address these issues and improve the delivery precision. These emerging strategies are further discussed in the future perspectives section.

### 3.4. Microparticles

MPs are spherical polymeric carriers ranging from 1 to 2 mm in diameter, specifically designed to encapsulate therapeutic agents. MPs are injectable formulations, offering ease of administration. The encapsulation capacity of MPs provides substantial advantages over conventional dosage forms, including enabling the delivery of otherwise challenging drugs, such as hydrophobic and high-molecular-weight compounds. Moreover, encapsulation within the polymeric matrix provides protection from external factors, such as oxidation and enzymatic degradation, thereby improving drug stability, prolonging bioavailability, and reducing the dosing frequency [35,36].

Biodegradable materials, such as PLGA, PCL, poly(glycolic acid), and their copolymers, are commonly used to fabricate MPs to achieve a controlled and sustained drug release. MPs exhibit a lower risk of burst release than fibers and SFs, ensuring more stable therapeutic concentrations over time [37].

Arestin^®^ (Bridgewater, NJ, USA) is a PLGA-based MP formulation containing minocycline hydrochloride (1 mg). It was approved by the USFDA in the early 2000s [38]. It maintains bactericidal drug levels (>1 μg/mL) in the GCF for up to 14 days, effectively targeting periodontal pathogens [39]. Arestin^®^ has been reported to significantly reduce PPD and improve CAL when used adjunctively with SRP. Periocline^®^(Takatsuki, Osaka, Japan), available in Japan, is another minocycline-containing MP designed to sustain antibacterial levels (>0.1 μg/mL) for 7 days. It has demonstrated clinical efficacy in reducing PPD when combined with SRP [40]. However, MP washout due to GCF flow and the need for repeated applications remain significant challenges for MP-based delivery systems. Therefore, improving drug retention within the periodontal pocket is a key area for further optimization.

Since the introduction of Arestin^®^ and Periocline^®^, various MP-based LDDSs have been investigated [41]. However, recent research has increasingly focused on NPs that offer enhanced precision and therapeutic control. However, they have not yet reached clinical applications.

## 4. Future Perspectives of LDDSs in Periodontal Therapy

### 4.1. Overview

Although enhancing conventional LDDSs to meet key criteria, such as sustained release, is fundamental, these features alone are insufficient for next-generation systems. Advances in the microbiology, particularly biofilm biology, have underscored the need to move beyond planktonic cell treatment and develop drug delivery strategies targeting the biofilm structure [42]. Biofilms are up to 1000-fold more resistant to drugs than planktonic bacteria due to their extracellular matrix (ECM), which acts as a barrier to drug penetration [43]. Even potent antimicrobials, including certain antibiotics and CHX, often fail to reach deeper biofilm layers, which limits their efficacy [44]. Therefore, developing LDDSs capable of enhancing drug permeability or disrupting the ECM is crucial.

Furthermore, the judicious use of conventional antibiotics has been increasingly emphasized in response to the global threat of antimicrobial resistance [45]. In this context, high-precision LDDSs, such as stimuli-responsive systems, offer a rational strategy by enabling a trigger-based drug release, thereby reducing antibiotic exposure. Moreover, recent advances in LDDS platforms have broadened the spectrum of deliverable agents beyond conventional antibiotics [12]. A growing number of studies now report the successful incorporation of alternative antimicrobial agents, such as natural extracts and antimicrobial peptides, into LDDSs. While traditional agents like minocycline are still widely used, these alternative compounds offer additional therapeutic benefits and may help mitigate antimicrobial resistance. Furthermore, certain nanoparticle formulations have demonstrated intrinsic antimicrobial activity, adding another functional dimension to LDDSs.

The following sections introduce novel LDDS platforms that meet these emerging requirements, including gel- and NP-based systems, as well as a distinct class of drug carriers, ionic liquids (ILs) (Figure 2). This is followed by an overview of DDSs designed to promote alveolar bone regeneration, which is the ultimate therapeutic goal of periodontal therapy.

### 4.2. Gels

Smart hydrogels have emerged as a promising strategy to overcome the limitations of conventional hydrogels and provide highly precise drug release profiles [46]. These advanced systems can respond to external stimuli, enabling more precise, controlled, and adaptive drug delivery tailored to the local periodontal environment. The following subsections introduce recent advances in gel-based LDDSs. Table 2 shows promising gel-based LDDSs, and Table 3 lists those currently under clinical investigation. Both tables are organized according to the types of external stimuli to which the systems respond, and they highlight the formulation components and functional characteristics of each approach. These tables serve as overviews to facilitate the understanding of the diverse strategies in gel-based LDDSs. A further contextual background and detailed discussion of each system are provided in the main text.

#### 4.2.1. Enhanced Retention Capability

Smart hydrogels are designed to undergo sol-to-gel transition in situ upon exposure to specific stimuli within the periodontal pocket. These systems are highly fluid in their sol phase, allowing for easy injection and close adaptation to irregular pocket geometry. Upon stimulation, they solidify into a gel, thereby enhancing retention at the target site and enabling a sustained and localized drug release.

Thermoresponsive hydrogels are the most studied strategy in this context. When combined with agents such as sodium β-glycerophosphate, chitosan exhibits temperature-triggered gelation and retains its natural mucoadhesive properties, thereby improving the residence time [47,48]. In animal models, thermoresponsive chitosan gels loaded with minocycline significantly reduced PPD and GI compared with conventional minocycline ointments [49]. Additionally, synthetic polymers, such as poloxamers (triblock copolymers), have been used to formulate thermoresponsive hydrogels [50,51].

Light-responsive hydrogels are another gelation mechanism. In the presence of photoinitiators, such as methacrylate, these hydrogels undergo rapid gelation upon exposure to ultraviolet or visible light, providing immediate solidification even in dynamic oral environments.

#### 4.2.2. On-Demand Drug Release

Smart hydrogels enable a stimuli-responsive on-demand drug release, enabling the delivery of therapeutic agents with spatial and temporal precision. Internal (e.g., pH and enzymes) or external (e.g., light and magnetic fields) stimuli can trigger structural changes, such as swelling and degradation, leading to a controlled drug release. This on-demand release reduces the drug dosage at the local site, facilitates the prolonged use of a single device, and may decrease the required administration frequency.

pH-responsive hydrogels take advantage of the acidic environment (pH < 6.0) in inflamed periodontal tissues. Under such conditions, chitosan becomes water-soluble, allowing it to dissolve, resulting in a site-specific drug release. In an animal model, chitosan gel loaded with plant-derived antimicrobial embelin effectively inhibited alveolar bone loss [52].

Light-responsive hydrogels undergo transformation via photothermal mechanisms in which photosensitizers absorb light energy and convert it to heat. This localized temperature increase induces volumetric changes in the swollen hydrogel network, thereby enabling a controlled drug release. A previous study showed that a hydrogel incorporating gold nanocages enabled the near-infrared-responsive release of tetracycline, significantly suppressing alveolar bone loss in a periodontitis mouse model [53].

#### 4.2.3. Limitations of Gel-Based LDDSs and Future Directions

Despite recent advancements, gel-based LDDSs still have several limitations. A burst drug release remains a major concern, even in smart systems [29]. Therefore, hybrid strategies, such as incorporating NPs into hydrogels, have been proposed to improve the release control. Additionally, despite the mechanical advantages of gels in pocket filling and drug retention, their antibiofilm efficacy is not yet well established. Although preclinical studies have shown promising results, clinical evidence remains limited, underscoring the need for further validation. Addressing these challenges through innovative hybrid systems and rigorous clinical research is crucial to fully realizing the therapeutic potential of smart hydrogels in periodontal therapy.

**Table 2 biomolecules-15-00903-t002:** Representative smart hydrogel-based LDDSs for periodontal treatment.

Hydrogel Matrix	Drug Incorporated	Study Design	Characteristic	Refs
*Thermoresponsive hydrogels*				
Chitosan/sodium ß-glycerophosphate	Minocycline	In vitro + in vivo (rat)	• Strong ntibacterial effects against *P. gingivalis.* • Significant reductions in PPD and GI compared with conventional minocycline ointment.	[49]
*Thermo- and pH-responsive hydrogels*				
Carboxymethyl–hexanoyl chitosan sodium ß-glycerophosphate	Naringin (flavonoid)	In vivo (mouse)	• Enhanced drug release under acidic conditions. • Significantly reduced alveolar bone loss and inflammatory cell infiltration, with downregulation of key inflammatory markers (TLR2, RAGE, and TNF-α).	[47]
Chitosan Quaternized chitosan sodium α,ß-glycerophosphate	Ornidazole	In vitro	• Releases drugs faster in acidic conditions (pH 4.0) and exhibits antibacterial activity against periodontal pathogens, including *P. gingivalis* and *P. intermedia.*	[54]
*pH-responsive hydrogels*				
Carboxymethyl chitosan Oxidized dextran	Embelin	In vitro + in vivo (rat)	• Controlled and targeted embelin release in inflamed periodontal sites. • Inhibits *P. gingivalis*, reduces inflammation, and enhances alveolar bone recovery.	[52]
*Light-responsive hydrogels*				
Gelatin methacrylate Gold nanobipyramids coated with mesoporous silica	Minocycline	In vitro	• NIR-triggered controllable minocycline release exhibits excellent antibacterial activity against *P. gingivalis.*	[55]
Polyvinyl alcohol Sodium alginate Carbon nanofiber	Icariin (flavonoid)	In vitro + in vivo (rat)	• NIR-activated photothermal drug release. • Significantly reduced alveolar bone loss upon NIR irradiation.	[56]
Poly(N-isopropyl-acrylamide-co-diethyl-aminoethyl methacrylate) Gold nanocages	Tetracycline	In vitro + in vivo (rat)	• Pulsed drug release upon NIR exposure, enhancing antibacterial efficacy. • Significantly inhibited alveolar bone loss compared with tetracycline gel in an animal model.	[53]
*ROS-responsive hydrogels*				
Phenylboronic acid–functionalized poly(ethylene imine) Oxidized dextran	Doxycycline Metformin	In vitro + in vivo (rat)	• Controlled drug release in response to ROS levels. • Exhibited remarkable antibacterial activity against *P. gingivalis* and promoted alveolar bone regeneration.	[57]
*pH- and ROS-responsive hydrogels*				
Carboxymethyl chitosan Dextran 4-Formylphenylboronic acid	Metal–organic framework of magnesium and gallic acid	In vitro + in vivo (rat)	• Controlled drug release in response to pH and ROS levels. • Enhances antibacterial activity against *P. gingivalis* and *A. actinomycetemcomitans* and promotes alveolar bone regeneration.	[58]
*Enzyme-responsive hydrogels*				
Polyethylene glycol–diacrylate MMP-8-sensitive peptide (CGPQG↓IWGQC *)	Minocycline Antibacterial peptide KSL (KKVVFKVKFK)	In vitro	• MMP-8-responsive drug release. • Effective inhibition of *P. gingivalis.*	[59]

ROS: reactive oxygen species; PPD: probing pocket depth; GI: gingival index; TLR2: toll-like receptor 2; RAGE: receptor for advanced glycosylation end products; TNF-α: tumor necrosis factor-α; MMP-8: matrix metalloproteinase-8; NIR: near-infrared. * CGPQG↓IWGQC: MMP-sensitive peptide sequence; the arrow (↓) indicates the specific cleavage site recognized by MMPs.

**Table 3 biomolecules-15-00903-t003:** Representative smart hydrogel-based LDDSs currently under clinical investigation.

Hydrogel Matrix	Drug Incorporated	Study Design	Characteristic	Refs
*Thermoresponsive hydrogels*				
Poloxamer 407 (Pluronic F127)	Green tea catechin extract	RCT (*n* = 30) Groups: Control: SRP + placebo gel Test: SRP + green tea catechin gel (once at baseline) Time: baseline and 1 month	Both groups showed improvements. However, the treatment group significantly outperformed the control group in all parameters (GI, PPD, and CAL)	[50]
*Thermo- and pH-responsive hydrogels*				
Pluronic F127 Carbopol P934	Curcumin	RCT (*n* = 20) Groups: Control: SRP Test: SRP + 2% curcumin in situ gel (once weekly for 3 weeks) Time: baseline and 1 month	Significant improvements in PPD and BOP were observed in the test group	[51]

RCT: randomized controlled trial; SRP: scaling and root planning; GI: gingival index; PPD: probing pocket depth; CAL: clinical attachment levels; BOP: bleeding on probing.

### 4.3. Nanoparticles

Recently, NPs have emerged as a promising platform for treating periodontitis. Besides the beneficial characteristics shared with MPs, such as prolonged drug bioavailability and the ability to provide a controlled and sustained release, NPs offer further advantages. Their high surface-area-to-volume ratio allows for a greater drug-loading capacity [60]. Moreover, their nanoscale size facilitates spatiotemporally precise drug delivery to hard-to-reach areas, such as dentinal tubules and intracellular spaces.

The antibiofilm capability is a particularly notable advantage of NPs [61]. With sizes typically ranging from 10 to 500 nm, NPs can penetrate biofilm channels and remain retained longer, thereby improving localized antimicrobial activity and facilitating biofilm disruption [62]. Additionally, some NPs are designed to degrade the biofilm ECM, whereas others possess intrinsic antimicrobial properties to enhance the antibiofilm capability and offer the potential for antibiotic-free therapies.

The following sections highlight the key NP types reported for their antibiofilm efficacy. Table 4 shows promising NP-based LDDSs and Table 5 lists those currently under clinical investigation. Both tables are organized by NP type and summarize the incorporated agents, formulation characteristics, and key therapeutic outcomes. They highlight diverse functional strategies, such as the photodynamic responsiveness, pH sensitivity, and enhanced retention, underscoring the potential of NPs in preclinical and translational periodontal therapy. Further background and detailed discussions are provided in the main text below.

#### 4.3.1. Metallic NPs

Metallic NPs, such as silver (AgNPs), gold (AuNPs), and platinum (PtNPs), have been extensively studied for periodontal applications due to their intrinsic broad-spectrum antimicrobial activity against periodontitis-associated pathogens. Their primary mechanism involves the direct disruption of bacterial cell membranes, with minimal risk of resistance development [63,64].

Although AgNPs may pose cytotoxic risks due to the release of free Ag^+^ ions, PtNPs are considered more biocompatible due to their chemical stability and corrosion resistance. Additionally, they demonstrated strong biofilm-disrupting activity [65,66]. Although AuNPs demonstrate excellent biocompatibility, their inherent antimicrobial activity is relatively weak. However, they have also been used as photothermal agents in photothermal therapy [67]. Upon near-infrared irradiation, AuNPs generated localized heating and released antimicrobial compounds, such as polyphenols, resulting in a synergistic reduction in biofilm mass and the suppression of alveolar bone loss in a ligature-induced rat periodontitis model [68].

Given their lack of degradability, a thorough evaluation of the cytotoxicity and environmental impact of metallic NPs is crucial.

#### 4.3.2. PLGA NPs

PLGA was approved by the USFDA as a standout biocompatible synthetic polymer. PLGA lacks intrinsic antimicrobial properties. Therefore, its antibacterial and antibiofilm effects rely on the encapsulated agents [69,70].

SspB adherence region (BAR) peptide-encapsulated PLGA NPs have been uniquely developed [71,72]. BAR-PLGA NPs exhibited the superior dose-dependent disruption of the established biofilms (IC₅₀ = 1.3 μM) compared with free peptides due to the BAR peptide’s ability to inhibit interactions between *P. gingivalis* and *Streptococcus gordonii* during biofilm formation. BAR-PLGA NP treatment significantly reduced alveolar bone loss in a murine periodontitis model, indicating its therapeutic potential.

Additionally, PLGA has been increasingly used in photodynamic therapy (PDT) for periodontitis to improve the retention and tissue penetration of photosensitizers such as methylene blue (MB) [73]. MB-loaded PLGA NPs have been reported to exhibit enhanced antibacterial and antibiofilm effects compared with free MB. A 3-month clinical trial in patients with chronic periodontitis demonstrated that PDT using MB-PLGA NPs as an adjunct to SRP significantly improved clinical outcomes [74].

#### 4.3.3. Chitosan NPs

Chitosan enhances NP retention and tissue penetration due to its natural mucoadhesive properties [75]. Additionally, its broad-spectrum antibacterial and antifungal activities have made chitosan widely applicable in the dental field [76]. As discussed in the Gels section, its stimuli-responsiveness enables sophisticated drug delivery. Therefore, chitosan NPs alone have demonstrated therapeutic utility due to these advantages [77]. However, chitosan is frequently used as a coating material to enhance the performance of other NPs, thereby improving their drug release profiles and pharmacokinetics. For example, chitosan combined with PLGA has been reported to impart antibacterial properties to PLGA and address its limitations, such as a burst drug release. Chitosan-coated PLGA NPs exhibited a more controlled release profile (sustained for up to 14 days) and demonstrated antibacterial and antibiofilm activities superior to unmodified PLGA NPs [78].

However, chitosan-based NPs tend to aggregate, thereby reducing the drug dispersibility and sustained release capability, which is a commonly reported challenge. Overcoming this limitation is expected to further facilitate the clinical application of chitosan NPs in periodontal therapy.

#### 4.3.4. Nanoliposomes

Nanoliposomes, spherical NPs composed of a phospholipid bilayer, are designed to mimic biological membranes in terms of their structure and function, offering superior biocompatibility and a slow-release drug profile [10]. The therapeutic potential of nanoliposomes has been demonstrated in a rat periodontitis model in which nanoliposomes delivery of minocycline hydrochloride improved the GI and PPD [79]. Furthermore, chitosan-coated nanoliposomes, designed to provide a pH-responsive drug release, loaded with doxycycline have been reported to effectively inhibit biofilm formation and alveolar bone loss in an experimental periodontitis model [80].

Tocosomes, another class of bilayer vesicles, exhibit distinct characteristics compared to conventional nanoliposomes. Due to their intrinsic antioxidant properties derived from tocopherol, tocosomes offer additional biological benefits, such as the modulation of oxidative stress. Although still in the early stages of investigation for periodontal therapy, their favorable biocompatibility and bioactivity suggest promising potential for future applications in LDDSs targeting inflammatory oral diseases [81,82].

#### 4.3.5. NP Composites: Association Between NPs and Scaffolds

Combining NPs with hydrogels is a rational strategy to overcome the limitations of each system [83,84,85]. NPs often suffer from formulation instability, which can lead to rapid clearance from the periodontal pocket [14]. Therefore, hydrogels have been investigated as potential scaffolds to enhance NP retention at the target site. Smart hydrogels are at the frontier of this field of study, providing highly precise drug delivery profiles through enhanced retention and stimulus-responsive release triggered by factors such as temperature, pH, or light, as discussed in previous sections. Conversely, even smart hydrogels are prone to burst drug release, thereby compromising the sustained therapeutic efficacy. Combining NPs with hydrogels can mitigate this issue and improve the controlled and sustained drug delivery profiles.

For example, a thermoresponsive hydrogel incorporating chitosan NPs loaded with a natural antimicrobial extract demonstrated a prolonged drug release with reduced burst effects, thereby maintaining the inhibition of ECM formation over an extended period [86]. Additionally, a pH-responsive hydrogel incorporating minocycline and zinc oxide (ZnO) NPs demonstrated a pH-sensitive slow release and outperformed both minocycline ointment and free ZnO NPs in antimicrobial and therapeutic efficacy [87].

These composite systems are still in the early stages of investigation, but their synergistic advantages offer promising potential for future clinical translation in periodontal therapy.

#### 4.3.6. Limitations of NP-Based LDDSs and Future Directions

Given the advantages of NP-based LDDSs, these systems have considerable potential for advancing periodontitis therapy. However, most NP research is still in the preclinical stage, primarily in vitro, with only a few formulations having advanced to a clinical evaluation. No NP-based products have yet reached the market for periodontal treatment.

Therefore, accelerating translational research is critical for bringing NP-based therapeutics into clinical application. Furthermore, high manufacturing costs and complex fabrication processes associated with NPs pose significant challenges to large-scale production, which must be addressed to enable broader clinical adoption.

**Table 4 biomolecules-15-00903-t004:** Representative NP-based LDDSs for periodontal treatment.

NPs	Drug Incorporated	Study Design	Characteristic	Refs
AgNPs	Ebselen	In vitro + in vivo (rat)	• AgNPs combined with ebselen showed a synergistic antibiofilm effect. • Effectively reduced alveolar bone resorption.	[88]
AuNPs	Epigallocatechin gallate (photosensitizer)	In vitro + in vivo (rat)	• Significantly inhibited bacterial biofilms upon NIR irradiation. • Reduced dental plaque biofilm by 87% and promoted alveolar bone regeneration.	[68]
PtNPs	-	In vitro	• Mouthwash with PtNP (10 μL) reduced biofilm by 70% and plaque by 60% in 2 h.	[66]
ZIF-8 NPs	Cerium ions	In vitro	• Sustained Zn^2+^ and Ce^3+^/Ce^4+^ provide strong antibacterial and anti-inflammatory effects.	[89]
ZIF-8 NPs	Minocycline	In vitro + in vivo (rat)	• Sustained antibacterial and anti-inflammatory effects. • Effective reduction of alveolar bone resorption.	[90]
PLGA NPs	Minocycline	In vitro + in vivo (rat)	• Exhibits controlled and continuous release of minocycline hydrochloride over an extended period (21 days). • Significantly inhibited inflammation and promoted periodontitis recovery.	[69]
PLGA NPs	Peptide (BAR)	In vitro + in vivo (mouse)	• Inhibition of *P. gingivalis* and *Streptococcus gordonii* interactions and biofilm formation, with dose-dependent disruption of established biofilms compared with free peptides. • Significant reduction in alveolar bone loss and GI.	[71]
PLGA NPs	MB (photosensitizer)	In vitro (human sample)	• Enhanced phototoxicity against dental plaque bacteria in the planktonic and biofilm phases compared with free MB.	[91]
Chitosan-modified PLGA NPs	Paclitaxel	In vitro	• Improved drug release profile with reduced initial burst release from PLGA NPs and pH-responsive release.	[78]
Nanoliposomes	Minocycline	In vivo (mouse)	• Improvement in GI and PD, with a reduction in the number of mononuclear and broken bone cells over 14, 28, and 56 days.	[79]
Chitosan-modified nanoliposomes	Doxycycline	In vitro + in vivo (rat)	• pH-responsive drug release with effective inhibition of biofilm formation and alveolar bone loss.	[80]
*Composite*				
Polydopamine NPs + chitosan/ß-glycerol phosphate gel	Antimicrobial peptides(Nal-P-113)	In vitro + in vivo (rat)	• Stable drug release for up to 13 days with the thermosensitive composite system. • 80% scavenging activity against *F. nucleatum* and *P. gingivalis.* • Reduction in alveolar bone loss and inhibition of inflammation in a rat model of periodontitis.	[85]
Chitosan NPs + Pluronic F127/hyaluronic acid gel	*Opuntia ficus-indica* extract	In vitro	• Prolonged drug release enabled by the thermosensitive composite system. • Inhibition of biofilm formation and reduction of exopolysaccharide production.	[86]

NPs: nanoparticles; PLGA: poly(lactide-co-glycolide); MB: methylene blue; GI: gingival index; PD: probing depth, NIR, near-infrared.

**Table 5 biomolecules-15-00903-t005:** Representative NPs-based LDDSs currently under clinical investigation.

NPs	Drug Incorporated	Treatment	Clinical Effectiveness	Refs
AgNPs	-	RCT (*n* = 30) Groups: Control: 0.2% CHX mouthwash Test: AgNPs mouthwash Time: baseline and 15 days	The AgNP mouthwash efficiently reduced PI, GI, and CRP levels in the GCF. However, it was not equivalent to the CHX mouthwash	[92]
PLGA NPs	Curcumin	RCT (*n* = 20) Groups: Control: SRP + empty NPs Test: SRP + PLGA/PLA NPs loaded with 50 μg of curcumin (once at baseline) Time: baseline, 3, 7, and 15 days	Both groups showed similar improvements in PPD, CAL, and BOP, with no additional benefit in bacterial elimination observed in the test group compared with the control group	[70]
PLGA NPs	20% doxycycline	RCT (*n* = 40) Groups: Control: FMUD + placebo NPs Test: FMUD + doxycycline NPs (once at baseline) Time: baseline, 1, 3, and 6 months	In deep pockets, test NPs significantly improved BOP at 3 and 6 months, PPD at 3 months, and CAL at 1 and 3 months. A higher percentage of sites with ≥2 mm PPD reduction and CAL gain was observed in the test group at 3 months	[93]
PLGA NPs	Methylene blue (photosensitizer)	RCT (*n* = 10) Groups: Control: US + SRP Test: US + SRP + aPDT with the NP (once at baseline) Time: baseline, 1 week, 1 month, and 3 months	The test group showed significantly greater improvement in the gingival bleeding index than the control group at 3 months	[74]
Chitosan-modified PLGA NPs	Indocyanine green (photosensitizer)	RCT (*n* = 40) Groups: Control: PDT Test: PDT with the NPs Time: immediately after treatment and at 1 week	No significant differences in PPD and BOP between the groups. However, the bacterial colony counts were significantly lower immediately after treatment in the test group	[94]

NPs: nanoparticles; CHX: chlorhexidine; RCT: randomized controlled trial; SRP: scaling and root planning; PLGA: poly(lactide-co-glycolide); PLA: poly(lactic acid); FMUD: full-mouth ultrasonic debridement: US: ultrasonic scaling; aPDT: antimicrobial photodynamic therapy; PDT: photodynamic therapy; PI: plaque index; GI: gingival index; CRP: C-reactive protein; GCF: gingival crevicular fluid; PPD: probing pocket depth; CAL: clinical attachment levels; BOP: bleeding on probing.

### 4.4. Ionic Liquids

ILs are salts composed entirely of ions that remain in a liquid state at or near room temperature. ILs, a novel class of drug carriers, are fundamentally distinct from conventional polymers. The exceptional tissue permeability of ILs is one of their most notable features, which is largely attributed to ion–lipid interactions with biological barriers [95,96]. Among ILs, the choline and geranic acid (CAGE)-based IL has demonstrated outstanding permeability, enabling the transdermal delivery of high-molecular-weight drugs, such as insulin [97]. This marks a significant advancement in the development of topical insulin formulations, which have traditionally been limited to injectable routes, offering a more patient-friendly alternative.

Our research group has explored the application of CAGE IL in periodontal therapy and reported its strong therapeutic potential [98,99] (Figure 3). CAGE IL has been reported to penetrate deeply into the periodontal biofilm structure. Moreover, it exhibits intrinsic antimicrobial activity by disrupting bacterial cell membranes, causing the death of periodontal pathogens and the subsequent collapse of the biofilm. Notably, this antibiofilm effect occurs within minutes. The topical application of CAGE IL significantly suppressed alveolar bone loss in a rat periodontitis model.

Beyond its antibiofilm effects, CAGE IL offers a unique clinical advantage due to its self-penetrating capability into periodontal pockets. In vivo studies in rats confirmed that topically applied CAGE IL rapidly permeates into the deep periodontium, eliminating the need for direct pocket insertion, an attribute that is not demonstrated by conventional LDDSs. This simplified, noninvasive delivery approach enhances patient compliance and broadens clinical utility. Additionally, CAGE IL offers practical benefits such as easy fabrication and low production costs, supporting its feasibility for mass production.

Future directions include leveraging CAGE IL’s drug carrier capabilities for dual-function delivery systems that integrate antimicrobial activity with the targeted delivery of additional therapeutics, such as anti-inflammatory drugs. Moreover, the accumulation of robust clinical data is essential for advancing CAGE IL-based formulations toward widespread clinical use in periodontal therapy.

### 4.5. LDDSs for Alveolar Bone Regeneration

The crucial aim of periodontal therapy is to increase the alveolar bone mass and restore the functional aspects of the periodontium. The triad of cells, scaffolds, and growth factors forms the cornerstone of successful regeneration in periodontal tissue engineering. LDDS-based approaches hold great promise in supporting this triad by enabling precisely designed drug delivery that accommodates the complex and time-consuming nature of tissue healing processes [100].

Table 6 presents recent LDDS-based strategies aimed at promoting alveolar bone regeneration. As this area of research remains in its early stages, the listed systems are currently limited to preclinical studies and have not yet reached clinical investigation. The table provides a concise overview of how these experimental approaches address existing limitations in periodontal regeneration. Further background information and detailed discussions of each system are provided in the main text.

Growth factors are key biological signals that regulate cell proliferation, differentiation, and migration. However, their clinical application is hindered by poor in vivo stability and limited retention at the target site. For instance, bone morphogenetic protein-2 (BMP-2) has a half-life of only 7 min. Even when delivered via traditional collagen scaffolds, BMP-2 is released in a burst manner, resulting in suboptimal regeneration outcomes and occasionally serious complications, such as ectopic bone formation [101]. A Nap-Phe-Phe-Tyr-OH-based hydrogel codelivering BMP-2 and stromal cell-derived factor-1 achieved a synchronous and sustained release for over a month, resulting in a bone volume fraction of 56.7% in a rat periodontal defect model, significantly outperforming conventional systems [102].

Scaffolds provide structural support for cell attachment and proliferation. They should exhibit high cellular affinity and biodegradability synchronized with the tissue regeneration process. Autografts are considered the gold standard due to their inherent osteogenic, osteoinductive, and osteoconductive properties. However, their clinical application is limited by donor site morbidity, postoperative pain, and restricted tissue availability. Allografts and xenografts are alternatives, but pose potential risks of immune rejection and pathogen transmission. Conversely, synthetic grafts, such as *β*-tricalcium phosphate, hydroxyapatite, and carbonate apatite, offer excellent biocompatibility and osteoconductivity. However, they lack inherent osteogenic and osteoinductive capabilities and often exhibit suboptimal resorption behavior. Therefore, LDDS-based scaffolds are being developed with programmable degradation profiles and architectures engineered to mimic the native ECM to enhance the cellular affinity and functional integration [100].

Besides structural support, scaffolds play a regulatory role in modulating the regenerative microenvironment. Since bone regeneration is a tightly orchestrated sequence of biological events, including inflammation, cell proliferation, ECM deposition, and remodeling, DDS strategies for periodontal regeneration must be multifunctional and phase-specific. For example, PLGA–lovastatin–chitosan–tetracycline NPs, engineered to sequentially release tetracycline (to manage early-stage infection) followed by lovastatin (to stimulate osteogenesis during the later phase), significantly promoted new bone formation in a canine periodontitis model [103].

Despite being in the developmental stage, the integration of multifunctional LDDSs into scaffold platforms represents a promising and transformative strategy in periodontal regenerative therapy, with the potential to achieve more predictable, efficient, and biologically coordinated healing outcomes.

**Table 6 biomolecules-15-00903-t006:** Representative LDDSs for alveolar bone regeneration.

LDDSs	Drug incorporated	Study design	Characteristic	Refs
Nap-Phe-Phe-Tyr-OH-based hydrogel	• Stromal cell-derived factor-1 • BMP-2	In vitro + in vivo (rat)	• Synchronous and sustained release of cell-derived factor-1 and BMP-2 over 35 days. • Achieved a bone volume fraction of 56.7%, indicating enhanced bone regeneration.	[102]
Chitosan-modified PLGA NPs + gelatin	• Tetracycline • Lovastatin	In vitro + in vivo (dog)	• Sequential release of tetracycline (infection control) and lovastatin (osteogenesis). • Significantly increased new bone formation in defects filled with the system.	[103]
Asymmetric membrane Aspirin-PLGA-NP/curcumin + collagen nanofibers	• Aspirin • Curcumin	In vitro + in vivo (dog)	• Burst release of aspirin followed by sustained release until day 30. • Complete new bone formation at 28 days, whereas no bone was found in the commercial membrane area.	[104]

BMP-2: bone morphogenetic protein-2; PLGA: poly(lactide-co-glycolide).

## 5. Conclusions

Advances in the understanding of the pathogenesis of periodontitis have increased interest among researchers and clinicians in developing novel therapeutic strategies aimed at improving clinical outcomes. LDDSs represent a transformative technological approach that offers precise dosing and overcomes the limitations of conventional treatment modalities. With recent advances in biomaterials and nanotechnology, the potential of LDDSs not only for effective infection control, but also for promoting alveolar bone regeneration has significantly broadened. Furthermore, in the context of antimicrobial resistance, LDDSs offer the advantage of reduced antibiotic dosages and enable the use of alternative antimicrobial agents. Targeting biofilm-specific structures and enhancing drug penetration into periodontal pockets are key strategies for future systems. Emerging technologies, such as ILs, smart hydrogels, and NP-based platforms, are leading this evolution.

Although many novel LDDSs remain in the experimental phase, continued interdisciplinary research and robust clinical validation are crucial to translating these innovations into practical periodontal therapies. LDDSs are poised to play a crucial role in the future of minimally invasive and precision-based periodontal care.

## Figures and Tables

**Figure 1 biomolecules-15-00903-f001:**
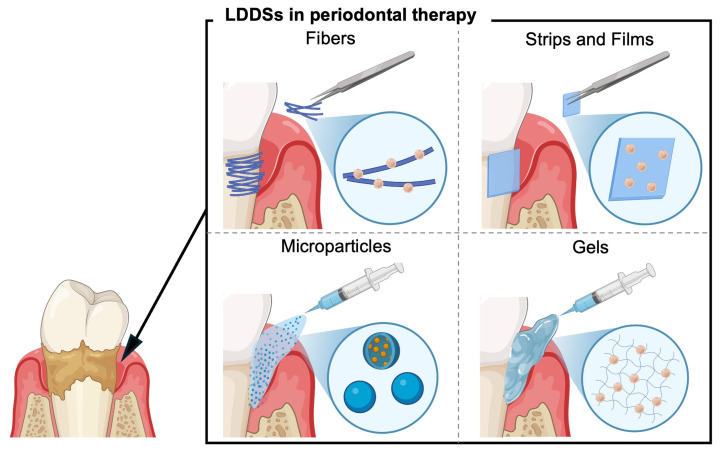
Schematic representation of clinically applied local drug delivery systems (LDDSs) used in periodontal therapy, including fibers, strips and films, microparticles, and gels.

**Figure 2 biomolecules-15-00903-f002:**
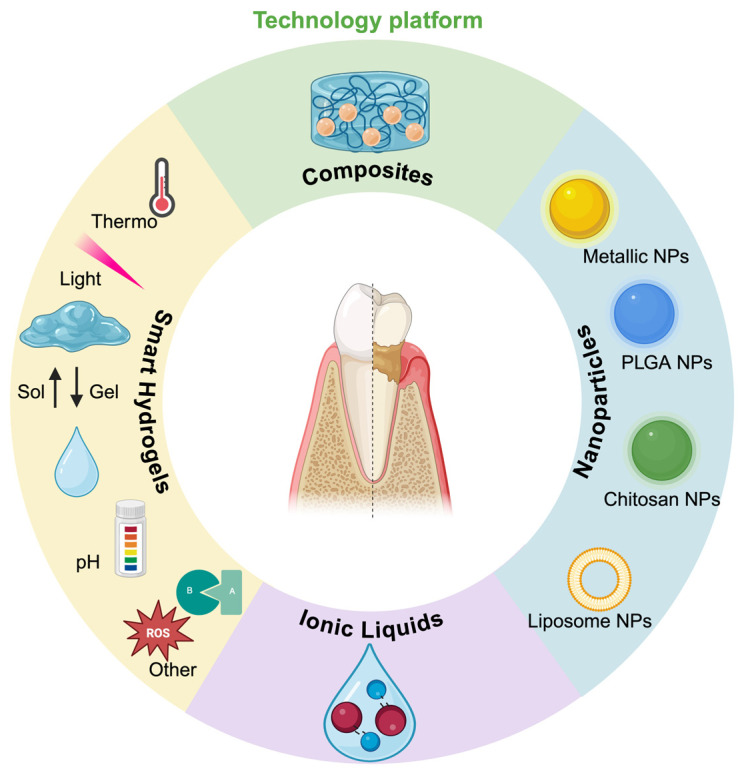
Schematic illustration of the emerging LDDS platforms for periodontal therapy.

**Figure 3 biomolecules-15-00903-f003:**
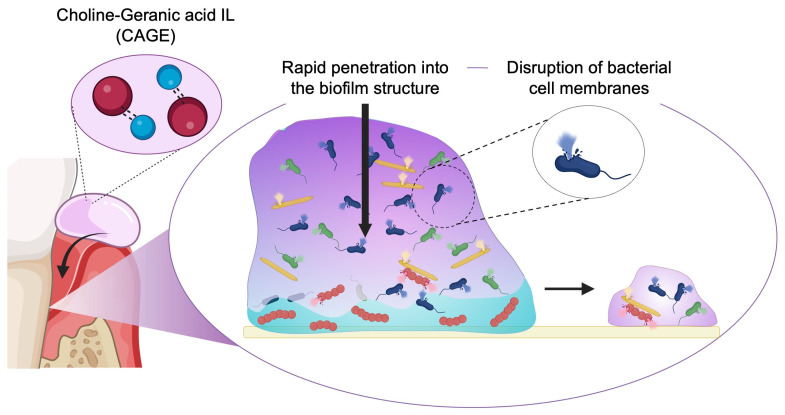
Schematic illustration of choline and geranic acid-based ionic liquid (CAGE IL) for periodontal treatment. The figure shows the topical application of CAGE IL to the gingiva, its self-penetration into periodontal pockets and biofilm structures, and its intrinsic antimicrobial activity via bacterial membrane disruption.

**Table 1 biomolecules-15-00903-t001:** Current status of clinically applied periodontal local drug delivery system (LDDS).

	Easy Application	Retention	Controlled and Sustained Drug Release	Biodegradability	Biocompatibility
Fibers	No	Yes	Yes	Partial	No
	Requires professional insertion	Structural retention	Sustained drug release over days	Depends on the material	Local tissue irritation
SFs	Partial	Yes	Yes	Yes	Yes
	Easier insertion, but requires clinical expertise	Mucoadhesive	Initial burst release followed by sustained drug release	Biodegradable polymers	Well-tolerated by periodontal tissues
Gels	Yes	No	Partial	Yes	Yes
	Injectable formulations	Rapid elimination	Initial burst release followed by sustained drug release	Biodegradable polymers	High biocompatibility
MPs	Yes	No	Yes	Yes	Yes
	Injectable formulations	Drug washout due to GCF flow	Sustained release with minimal burst risk	Biodegradable polymers	Well-tolerated by periodontal tissues

SFs: strips and films; MPs: microparticles; GCF: gingival crevicular fluid.

## Data Availability

No new data were created or analyzed in this study.

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
