# Peer review of "Advances in Local Drug Delivery for Periodontal Treatment: Present Strategies and Future Directions"

_biomolecules, 2025, doi:10.3390/biom15060903_

Round 1
Reviewer 1 Report
Comments and Suggestions for Authors
This review article by Mayoka has focused on local delivery strategies for Periodontitis treatment. This is an extensive and through review contains advanced on this field, perspective and expert options. The claims and statement are supported by adequate references. Without further revision, I recommend for acceptance for publication as it is.
Author Response
We sincerely thank the reviewer for the positive and encouraging comments on our manuscript. We are grateful for your recognition of its relevance and comprehensiveness, and we truly appreciate your recommendation for acceptance without further revision.
Reviewer 2 Report
Comments and Suggestions for Authors
The paper “
Advances in Local Drug Delivery for Periodontal Treatment: Present Strategies and Future Directions”
conducted by Mayuka Nakajima et al. provide very important informations regarding strategies used in periodontal therapy, highlighting existing active ingredient delivery systems as well as those in clinical trials.
The basic question refers to the therapeutic strategies that aim at the optimal approach in periodontitis, based on the advances obtained in understanding the pathogenesis of this disease.
The paper presents the topical systems used in periodontal therapy, both as vehicles of various active principles with antimicrobial, anti-inflammatory, antioxidant effects, but also as structural elements to promote alveolar bone regeneration.
Conventional and modern delivery systems, based on smart materials, such as nanomaterials and biotechnologies, are presented. The paper highlights the advantages and limitations of each system and presents concrete examples from the specialized literature. The paper is clearly developed, presenting the basic elements of each delivery system, which favors the understanding and usefulness of the information.
I was pleased to note the analytical, critical presentation, emphasizing the quality of the presentation, not the quantity. I congratulate the authors and confidently recommend the publication in this form.
The tables and figures are relevant and present the data accurately, being easy to interpret and understand.
The references are recent and relevant to the information presented.
Author Response
We would like to thank the reviewer for their thorough and insightful evaluation of our manuscript. It is encouraging to hear that our review was found to be comprehensive, logically organized, and clearly written. We deeply appreciate your positive assessment and recommendation for publication.
Reviewer 3 Report
Comments and Suggestions for Authors
The review presented a comprehensive overview of locally administrated DDSs in periodontal therapy, where in the first sections the currently available products were detailed while in the second part the potential future candidates were demonstrated.
Additionally, the review discussed the challenges and future perspectives of these local DDSs. The work is well-structured; the tables are informative.
I would like to make the following comments and suggestions regarding the work:
- Table 1: in my opinion, yes/no or partial is not understandable in the case of biocompatibility and biodegradability (two different things). Other observation, NPs are not indicated here.
- Chapter 3.2: in the title, "Matrix System" could apply to gels also, in my opinion, the title as „Strips and Films” would be better.
- It would be better to write more details about tables 2-6 in the text.
- Table 2: Authors would not indicate glycerol as a matrix component, since it usually only indicates polymers. What does (CGPQG↓IWGQC) mean in the table?
- Table 3: Is the system really pH-responsive in the 2nd example? The one before it also contains Carbopol, it was not considered as, while in the second one it is. Authors would not list PEG 400 and triethanolamine in the latter, since in the other places you only listed polymers.
- It would be worth giving a summary of the locally applied active ingredients.
- In Figure 2, Authors also mention composites, but they do not make such in-depth assessments as in the case of gels, nanoparticles, or ionic liquids. Please elaborate on these systems as well.
Reviewer 4 Report
Comments and Suggestions for Authors
Dear Authors;
Re: [Manuscript ID: biomolecules-3638801]
Title: "Advances in Local Drug Delivery for Periodontal Treatment:
Present Strategies and Future Directions"
In this review article you aimed to provide to readership a comprehensive overview of the current and emerging DDS strategies in periodontal therapy, focusing on their applications in infection management and bone regeneration and discussing their limitations and prospects for clinical translation.
The manuscript is very well-written, scientifically sound, and well illustrated. I only have few concerns as listed below.
- When checking Google Scholar, we get around 20,000 REVIEW articles on the topic of your paper. This means you need to clearly highlight novel / unique aspects of your manuscript.
- Please check the word "planning"! in Line 52.
- Paragraph 2 under the section Two have no References.
- Please double-check the statement: "... with reference to their clinical application, ..." (Line 93).
- Figure sources not mentioned (any copyright issues?).
- Table 1 have no References.
- In Line 199 consider using "applications" instead of "application".
- In Line 316 please correct the phrase: "nondegradability".
- Same with "preexisting" in Line 324.
- Under the section Liposomes: Definition of liposomes is wrong. It should be seriously noted that liposomes are micrometric - their nanometric versions are called NANOLIPOSOMES. Please consult published articles on the topic and correct this section accordingly.
- Consider to include a brief description on the potential applications and future perspectives of the most recent, most novel Drug Delivery System, Tocosome.
- In Line 452 please correct: "programable".
Thank you and good luck with your future research and studies.
Round 2
Reviewer 3 Report
Comments and Suggestions for Authors
The authors complied with all requests.
Reviewer 4 Report
Comments and Suggestions for Authors
Dear Authors,
Re: [Manuscript ID: biomolecules-3638801]
Title: "Advances in Local Drug Delivery for Periodontal Treatment: Present Strategies and Future Directions"
Thank you for the revised version of your manuscript.